# Experience of Managing Suspected Placenta Accreta Spectrum with or without Internal Iliac Artery Balloon Occlusion in Two Lithuanian University Hospitals

**DOI:** 10.3390/medicina57040345

**Published:** 2021-04-02

**Authors:** Egle Savukyne, Laura Liubiniene, Zita Strelcoviene, Ruta Jolanta Nadisauskiene, Edita Vaboliene, Egle Machtejeviene, Rytis Kaupas, Dalia Lauzikiene

**Affiliations:** 1Department of Obstetrics and Gynaecology, Hospital of Lithuanian University of Health Sciences, Kauno Klinikos, Eiveniu st. 2, 50161 Kaunas, Lithuania; zita.strelcoviene@lsmu.lt (Z.S.); ruta.nadisauskiene@lsmu.lt (R.J.N.); edita.abromaviciute@gmail.com (E.V.); egle.machtejeviene@lsmuni.lt (E.M.); 2Department of Obstetrics and Gynaecology, Medical Academy, Lithuanian University of Health Sciences, A.Mickevicius st. 7, 44307 Kaunas, Lithuania; rytis.kaupas@lsmu.lt; 3Department of Interventional Radiology, Hospital of Lithuanian University of Health Sciences, Kauno Klinikos, Eiveniu st. 2, 50161 Kaunas, Lithuania; 4Department of Obstetrics and Gynaecology, Vilnius University Hospital Santaros Klinikos, Santariskiu st. 2, 08661 Vilnius, Lithuania; dalia.lauzikiene@santa.lt

**Keywords:** internal iliac artery balloon, placenta accreta spectrum, placenta previa, cesarean hysterectomy, postpartum hemorrhage

## Abstract

*Background and objectives*: Placenta previa and placenta accreta spectrum are considered major causes of massive postpartum hemorrhage. Objective: To determine whether the placement of an occlusion balloon catheter in the internal iliac artery could reduce bleeding and other related complications during cesarean delivery in patients with placenta previa and placenta accreta spectrum. *Materials and Methods*: A retrospective analysis was conducted at two tertiary obstetric units of Lithuania. From January 2016 to November 2019 patients with placenta previa and antenatally suspected invasive placenta were included in the intervention group and underwent cesarean delivery with endovascular procedure. From January 2014 to December 2015 patients with placenta previa and suspected placenta accreta spectrum were included in the non-intervention group. The primary outcomes were reduction in intraoperative blood loss and transfusion volumes in the intervention group. Secondary outcomes were the incidence of hysterectomy and maternal complications. *Results*: Nineteen patients underwent cesarean delivery with preoperative endovascular procedure, and 47 women underwent elective cesarean delivery. The median intraoperative blood loss (1000 (400–4500) mL vs. 1000 (400–5000) mL; *p* = 0.616) and the need for red blood cell transfusion during operation (26% vs. 23%; *p* = 0.517) did not differ significantly between the patients groups. Seven patients in the intervention group and two patients in the non-intervention group underwent perioperative hysterectomy (*p* = 0.002). None of the patients had complications related to the endovascular procedure. *Conclusion*: The use of intermittent balloon occlusion catheter in patients with placenta pathology is a safe method but does not significantly reduce intraoperative blood loss during cesarean delivery.

## 1. Introduction

Placenta previa is an obstetric condition in which the internal cervical os is covered partially or totally by placental tissue. This pathology is a significant risk factor for intrapartum and postpartum hemorrhage and placenta accreta spectrum (PAS) [1,2]. This spectrum of abnormal invasion of the chorionic villi includes placenta accreta, increta, and percreta. Uncontrolled massive postpartum hemorrhage caused by placenta previa or PAS may lead to blood transfusion, hysterectomy, infertility, admission to the intensive care unit, extended hospital stay, or even maternal death [3,4]. The incidence of abnormal invasion of placental tissue is increasing. One reason for this is the increase in rate of cesarean deliveries [5]. However, in 3% of patients with placenta previa and no previous cesarean delivery, placenta accreta and its variants occur [6]. The risk of PAS drastically increases with the combination of placenta previa and last cesarean section (CS), especially when the placenta is implanted over a prior cesarean scar [7]. 

Antenatal diagnosis of PAS facilitates improved outcomes for mothers and neonates due to planned delivery at a tertiary maternal care facility by a multidisciplinary team [6,8]. The primary diagnostic method of placenta pathology is ultrasonography performed by experts [6]. Magnetic resonance imaging might be a useful tool for diagnosing difficult cases such as posterior placenta previa or evaluating invasion depth [9].

Placenta previa with PAS is a challenge for surgeons. Standard management of this clinical situation involves peripartum hysterectomy with the placenta left in situ [10]. Hysterectomy causes loss of fertility, so the desire to preserve the uterus and its function requires the use of alternative methods such as uterine compression sutures with intrauterine balloon tamponade, pelvic artery ligation, spiral suturing of the lower uterine segment or preoperative prophylactic balloon occlusion of intravascular pelvic arteries (abdominal aorta, common iliac artery, or internal iliac artery).

Prophylactic placement of artery balloons could reduce the incidence of hysterectomy and preserve fertility [11,12]. When the intra-arterial balloons are inflated to occlude the arteries, they do not arrest blood flow completely, but decrease the pulse pressure distal to the site of occlusion. Occlusion balloon catheter (OBC) placement aims to reduce the rate of blood loss during cesarean delivery and perioperative hysterectomy. However, studies have presented conflicting results regarding the usefulness of prophylactic balloon placement in the pelvic arteries. 

In January 2016, prophylactic placement of internal iliac artery balloon catheters was considered for patients with PAS and placenta previa in Lithuania. We evaluated the efficacy of preoperative placement of OBCs in the internal iliac arteries to reduce intraoperative blood loss during CS for women with placenta previa and PAS.

## 2. Materials and Methods

The study involved retrospective analysis of cases from two tertiary care level university hospitals in Lithuania. We collected clinical data of 66 patients with placenta previa who were prenatally diagnosed with or suspected of having PAS, based on ultrasound or magnetic resonance imaging data from electronic medical record databases and medical reports at two tertiary university obstetrics units (Lithuanian University of Health Sciences Kauno klinikos and Vilnius Univeristy Hospital Santaros klinikos), which are referral centers for high-risk obstetric patients in Lithuania. The keywords used for searching electronic medical record databases were placenta accreta, placenta increta, placenta percreta, and placenta previa. From January 2016, patients with prenatal suspicion of PAS underwent prophylactic placement of iliac occlusion balloons (intervention group (IG)). Patients who had placenta previa and PAS between January 2014 and December 2015 served as the non-intervention (NIG) group. We analyzed the data of only elective CSs. These numbers have been reported in Figure 1.

All patients (IG and NIG) underwent ultrasonography by experienced obstetrician-gynecologists to evaluate sonographic PAS markers and risk level. The typical 2-D gray-scale and color Doppler ultrasound findings were assessed for the presence of more than one of the signs: (1) Loss of hypoechogenic area between the uterus and placenta; (2) myometrial thickness <1 mm; (3) interruption of the uterine serosa-bladder wall interface; (4) abnormal placental lacunae with high-velocity flow >15 cm/s; (5) uterovesical hypervascularity, i.e., increased vascularity of the uterine serosa-bladder wall interface; and (6) irregular intraplacental vascularization [9]. Magnetic resonance imaging was not routinely performed. All IG patients underwent a CS, with only prophylactic internal iliac artery balloon placement. Those with placenta previa and suspicion of abnormal invasion of chorionic villi were prepared for blood transfusion, and blood and Rh factor blood tests were performed. The necessary units of mass of red blood cells were reserved. All patients received general anesthesia during cesarean delivery. Histopathological examination of the placenta was performed when experienced surgeons observed abnormal placentation signs during cesarean operation.

Patients suspected of having PAS underwent prophylactic perioperative internal iliac artery balloon occlusion on the day of delivery. The planned delivery day was decided on a case-by-case basis depending on contractile activity and bleeding after multidisciplinary counseling. Before the elective CS, the patients were taken to the interventional radiology department. A senior interventional radiologist performed the balloon occlusion procedure. During this procedure, the bilateral femoral arteries were punctured under local anesthesia. Balloon-tipped catheters were placed in both internal iliac arteries. After that, test inflation of the balloons was performed. The balloons were briefly inflated under minimal fluoroscopic guidance, and contrast material was injected to verify the volume of contrast required for complete occlusion of the internal iliac arteries. The catheters were securely taped to the skin, and patients were transported to the obstetrics operating theater for elective CS. After cord clamping, the balloons were inflated to reduce the blood supply to the uterus. If hemostasis was achieved, the balloons were deflated. Intra-arterial catheters were removed within 2 h after the cesarean operation. All women were managed by a multidisciplinary team (senior obstetrician-gynecologist, senior anesthesiologist, gynecologist-oncologist, neonatologist, and senior interventional radiologist).

The study’s primary goal was to evaluate intraoperative blood loss and the amount of blood products transfused in the IG and NIG. We estimated blood loss by measuring suction drainage and weighing surgical sponges and swabs. The secondary goal was to evaluate the incidence of hysterectomy in both groups, lengths of hospital and intensive care unit stay, and maternal complications, including adult respiratory distress syndrome, renal failure, ischemic liver and disseminated intravascular coagulation, or maternal death.

### 2.1. Statistical Analysis

Categorical variables were reported as numbers and percentages. Descriptive data were presented as median and range. The IG and NIG were compared using the Mann–Whitney U test for continuous variables and chi-squared test for categorical variables. A value of *p* ≤ 0.05 was considered statistically significant. Calculations were performed using IBM SPSS Statistics v27 (IBM Corp, Armonk, NY, USA). 

### 2.2. Ethical Approval

Ethical approval (Protocol number BEC-LSMU(R-09)) was obtained from the Bioethics Centre, Lithuanian University of Health Sciences, on 1 October 2019.

## 3. Results

The IG consisted of 19 patients with perioperative prophylactic internal iliac artery balloon occlusion, and the NIG, 47 patients with no intervention procedure. Both groups contained patients with placenta previa and suspected PAS. The demographic and obstetric characteristics were similar between groups, except that the prevalence of previous cesarean deliveries was higher in the IG (*p* = 0.012). All patients in the IG had at least one risk factor for PAS or placenta previa. The anterior previa location of the placenta was more common than the posterior previa location in the IG (Table 1).

Maternal and neonatal outcomes are presented in Table 2. 

We did not find any significant difference between the IG and NIG in terms of median perioperative blood loss (1000 (400–4000) vs. 1000 (400–5000) mL; *p* = 0.616). Furthermore, cases with histologically confirmed placenta accreta, increta, and percreta were analyzed separately. We did not find any difference in estimated blood loss (EBL) between the groups (Table 3).

The requirement of packed red blood cells and other blood components (transfusion of platelets, fresh frozen plasma, and cryoprecipitate) during the operation was not significantly different. Patients in the IG underwent perioperative hysterectomy, statistically and significantly more often than those in the NIG (*p* = 0.002). One patient in the IG had a partial excision of the uterus with placenta and uterus reconstruction. The reoperation rate was 3.03%. Two patients (one patient from each group) underwent relaparotomy after CS, and one woman in the IG underwent uterine artery embolization due to postoperative bleeding.

All patients were monitored after the operation in the ICU because all CSs were performed under general anesthesia. There was no difference in the extended stay in the ICU or postoperative ward. None of the women had complications related to hemorrhage (adult respiratory distress syndrome, renal failure, ischemic liver, or disseminated intravascular coagulation) or the endovascular procedure (hematoma at the puncture site, arterial thrombosis, or femoral nerve ischemic injury). Neonatal outcomes were not significantly different between the two groups.

## 4. Discussion

In this retrospective study, we did not observe reduced blood loss in patients with placenta previa and PAS with internal iliac arterial balloon occlusion. The main objective of the management of placenta previa and PAS is to reduce blood loss during cesarean delivery. The temporary placement of OBCs is a preventive measure to control hemorrhage in these cases. Although the latest reviews have shown controversial results, some studies have confirmed that this preventive procedure reduced perioperative blood loss, blood transfusion rate, and incidence of cesarean hysterectomy and improved visualization of the operating field [13,14,15,16]. However, these studies are limited by their retrospective nature and nonrandomized design. One large systematic review and meta-analysis showed that endovascular intervention was associated with reduced blood loss only for patients who underwent prophylactic balloon occlusion of the abdominal aorta [12]. Other studies did not show any benefit of balloon occlusion, including two randomized controlled trials on OBCs [17,18,19]. Salim et al. [18] examined patients with suspected placenta accreta who did or did not undergo balloon occlusion and found no changes between the groups. Furthermore, 15% of the patients experienced interventional radiology-related complications. None of our patients experienced complications during the balloon occlusion procedure. This could be due to the small number of cases in our IG or the short period from the perioperative internal iliac artery balloon insertion to displacement. Yu et al. [19] recently published data from a trial involving women with placenta previa: the trial involved 16 women who underwent prophylactic endovascular internal iliac artery procedures and 20 who were assigned to the control group; there was no reduction in the rate of perioperative hemorrhage or transfusion in the intervention group. Our study results also did not show any significant benefit of perioperative balloon occlusion in terms of EBL, need for blood transfusion, incidence of hysterectomy, and postoperative duration of hospitalization.

On the other hand, the safety of interventional radiology procedures in pregnancy is also a significant concern. However, several studies reported procedure-related complications: iliac or femoral artery thrombosis or rupture, ischemic nerve injury, or gluteal muscular necrosis [20,21,22,23]. One study reported possible causes related to the failure of the OBC procedure [17]. First, balloon occlusion may temporarily stop the blood supply via uterine arteries, but collateral vascularization is intense in the pregnant uterus, more so in the presence of an invasive placenta [24]. The lack of improved outcomes with endovascular procedures may be attributed to the marked opening of collateral circulation within the pelvis, precisely after inflation of intravascular balloons. This leads to increased blood flow to the uterine and cervical vessels and may result in massive blood loss.

In addition, balloon migration can occur in some cases [25]. In our study, it may be one of the causes of insufficient balloon occlusion of the arteries, since the placement of OBCs and CSs were performed in different departments. After patient admission to the obstetrics operative theater, we did not verify the exact balloon location, so the possibility of balloon migration cannot be ruled out. Finally, the main reason behind the lack of significant benefits of interventional procedures may be the heterogeneity of PAS. Cali et al. [26] showed the usefulness of OBC placement in cases of placenta percreta but not in those of accreta and increta. Meanwhile, our study did not show any significant difference between the placenta’s histological type and perioperative bleeding.

International organizations such as International Federation of Gynecology and Obstetrics (FIGO), Royal College of Obstetricians and Gynecologists (RCOG), and American College of Obstetricians and Gynecologists (ACOG) do not recommend the routine use of prophylactic pelvic OBCs for suspected PAS or placenta previa. There is a lack of evidence-based data on the safety and efficacy of prophylactic balloon occlusion. They suggest evaluating the risk/benefit ratio before the procedure for each case, except for patients who refuse donor blood transfusion [27,28,29]. Hence, prospective multicenter, randomized, and well-controlled studies are required to evaluate the efficacy and safety of pelvic artery balloon occlusion procedures.

The main limitations of the current study are its retrospective nature and small sample size. Therefore, selection and information biases may exist because of the absence of randomization. The IG and NIG characteristics were not well balanced, because the majority of the patients in the IG compared to NIG had a previous CS (84% vs. 47%), anterior placental location (79% vs. 49%), and cesarean hysterectomy (37% vs. 4%). Given the risk of massive bleeding during the operation, especially in cases of placenta previa anterior when the placenta is implanted over the prior cesarean scar, OBC placement was recommended for the IG. From January 2016, patients were selected to undergo OBC placement after a thorough assessment of risk factors, expert ultrasonography evaluation, placenta localization, the necessity to save fertility, and decision of the multidisciplinary team.

## 5. Conclusions

Our study showed that prophylactic internal iliac artery balloon occlusion during cesarean delivery is a safe procedure for both patients and newborns. We did not find any benefits of OBC placement in controlling bleeding during laparotomy in patients with placenta previa complicated by PAS. However, the data should be evaluated critically, because the intervention and non-intervention groups are markedly different according to previous caesarean deliveries, anterior placenta previa, and cesarean hysterectomy. For the accuracy of the data, it would be useful to continue the study in a prospective manner to evaluate the influence of internal iliac artery occlusion with balloon catheters and blood loss.

## Figures and Tables

**Figure 1 medicina-57-00345-f001:**
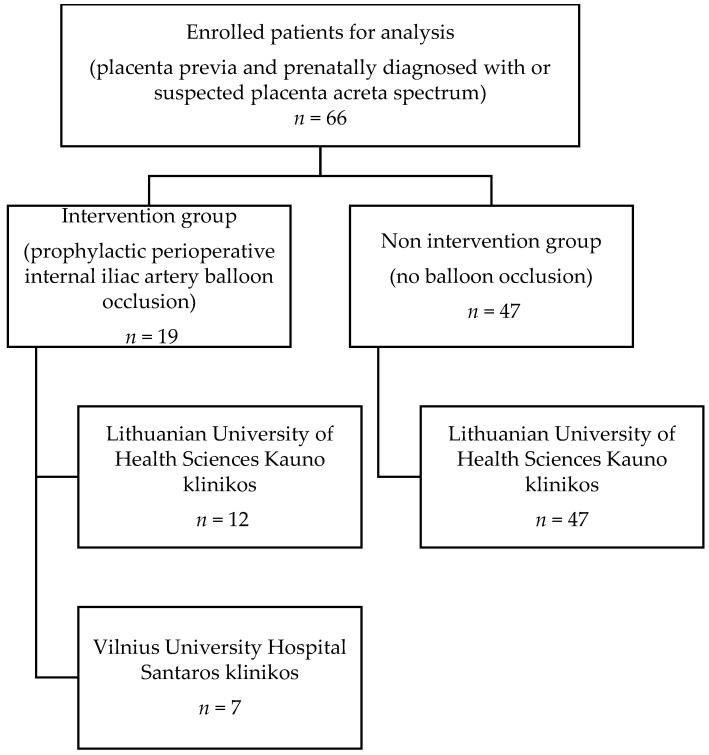
Flow diagram of cases selected for analysis. Study involved patients (*n* = 66) from two tertiary Lithuanian hospitals with confirmed placenta previa and with suspected or diagnosed with placenta accreta spectrum.

**Table 1 medicina-57-00345-t001:** Demographic and obstetric characteristics of the patients.

Demographic and Obstetric Characteristics	Intervention Group (*n* = 19)	Non-Intervention Group (*n* = 47)	*p*-Value
Maternal age (years)	33 (27–40)	34 (25–43)	0.594
Gravidity	3 (2–5)	3 (1–11)	0.691
Parity	2 (1–5)	2 (1–9)	0.888
Gestational age (weeks)	37 (32–38)	37 (30–40)	0.111
Patients with previous cesarean deliveries none 1 2 3 or more	16/19 (84%) 3/19 9/19 6/19 1/19	22/47 (47%) 25/47 11/47 5/47 6/47	0.012
Another risk factor (uterine curettage, abortion, manual removal of placenta, assisted reproduction pregnancy)	7/19 (37%)	16/47 (34%)	0.829
Placenta previa location			0.026
Anterior	15/19 (79%)	23/47 (49%)	
Posterior	4/19 (21%)	24/47 (51%)	

Data are shown as median (min–max value) or number (%).

**Table 2 medicina-57-00345-t002:** Maternal and neonatal outcomes.

Outcomes	Intervention Group (*n* = 19)	Non-Intervention Group (*n* = 47)	*p*-Value
Maternal outcomes			
Total EBL	1000 (400–6500)	1000 (400–5000)	0.637
Massive hemorrhage (≥1500 mL)	6 (32%)	15 (32%)	0.979
Transfusion of packed red blood cells	5 (26%)	11 (23%)	0.517
Units of packed red blood cells transfused	0 (0–16)	0 (0–5)	0.699
Transfusion of platelet, fresh frozen plasma, or cryoprecipitate	4 (21%)	5 (10.6%)	0.267
Uterine artery embolization after an operation	1 (0.05%)	0	-
Hysterectomy	7 (37%)	2 (4%)	0.002
Relaparotomy	1 (5.26%)	1 (2.13%)	-
Intensive care unit admission (≥2 days)	3 (16%)	3 (6%)	0.344
Postoperative hospital stay	6 (2–13)	6 (3–15)	0.657
Neonatal outcomes			
APGAR score at 1 min	8 (5–9)	8 (5–10)	0.507
APGAR score at 5 min	9 (6–10)	9 (7–10)	0.882

Data are shown as median (min–max) or number (%). (-)—not calculated due to lack of data.

**Table 3 medicina-57-00345-t003:** Association between placenta histology type and estimated blood loss (EBL).

	Intervention Group (*n* = 16)		Non-Intervention Group (*n* = 19)		*p*-Value
	Histology Gype	Median of EBL (Min–Max)	Histology Type	Median of EBL (Min–Max)	
Placenta accreta	5 (31%)	600 (500–1500)	7 (37%)	1600 (400–3500)	0.202
Placenta increta	7 (44%)	1000 (400–4000)	9 (47%)	1800 (500–4700)	0.350
Placenta percreta	4 (25%)	1750 (1000–6500)	3 (19%)	1700 (1000–5000)	1
Total	16	1000 (400–6500)	19	1700 (400–5000)	0.361

Data are shown as median (min–max) or number (%). EBL—estimated blood loss.

## Data Availability

The data presented in this study are available on request from the corresponding author.

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
