# Peer review of "Experience of Managing Suspected Placenta Accreta Spectrum with or without Internal Iliac Artery Balloon Occlusion in Two Lithuanian University Hospitals"

_medicina, 2021, doi:10.3390/medicina57040345_

Round 1
Reviewer 1 Report
The quality of the article has improved after the changes made according to the suggestions of the reviewers.
Author Response
Reviewer No.1:
The quality of the article has improved after the changes made according to the suggestions of the reviewers.
We really appreciate the careful reading and suggestions of the reviewers, which have helped to improve the manuscript.

Reviewer 2 Report
I have read the manuscript and I am satisfied with it.
The is one small correction for the second sentence in the "Conclusions" section on page 7. This sentence should not start with "And". Hence, simply remove "And". The sentence should be,
"We did not find any benefits................................"
Author Response
Reviewer No.2:
I have read the manuscript and I am satisfied with it.
The is one small correction for the second sentence in the "Conclusions" section on page 7. This sentence should not start with "And". Hence, simply remove "And". The sentence should be,
"We did not find any benefits................................"
Thank you for the valuable adjustment. Corrected as proposed. (Line 243).
Moderate English changes required.
The grammar is corrected by professional Editing. The certificate is attached.
